# Position: Federated Learning and Continual Learning researchers should compare notes

## Abstract

Federated Learning and Continual Learning are two learning paradigms where there are constraints on data access. In Continual Learning the constraints are temporal: the model is trained using different data over time. In Federated Learning the constraints are spatial: the model is trained using different data from different clients. A major problem in both paradigms occurs when access to the data partitions is heterogeneous. This can cause difficulties in converging to a well performing global model.

In this position paper we argue that the learning paradigms, data problems and difficulties in model training between Federated Learning and Continual Learning are the same problem along different axes (spatially and temporally respectively). We formalise this under the umbrella term of "Partitioned Learning". We highlight specifically how heterogeneity across partitions of data corresponds with types of failure mode and how this affects the corresponding strategies to mitigate failure. Given this, it is not surprising to see the emergence of similar strategies for failure mitigation, often, but not always, without referencing the other field. Our position is that these two scientific fields with such stark mathematical symmetries should collaborate in order to maximise research progress. Further progress could be made on both sides if researchers compared notes more frequently and considered more intentional collaborations.

## 1. Introduction

Federated and Continual Learning are both paradigms where there are constraints on the data available for model training. Data access could be spatially constrained: where the global training data is not all available in the same place and is instead spread across different "clients" (for privacy preservation); this setting is called "Federated Learning" (McMahan et al., 2016). Data access could also be temporally constrained, where data availability varies over time (the model could be trained on new data every day following collection); this setting is called "Continual Learning" (Wang et al., 2024). The third, rarer case is when data access is both spatially and temporally constrained, where data is spread across separate clients additionally get new data over time (Criado et al., 2022). This more niche setting is called "Federated Continual Learning" and is currently the main overlap of Federated Learning and Continual Learning literature.

> **Position: Federated and Continual Learning can be understood as analogous problems along different axes. They can both be considered as *Risk Minimisation over Partitioned Distributions*, where the partitions are in space or time respectively. Given this symmetry, it is not surprising that there is evidence that researchers on both sides are naturally converging on similar solutions for these failure modes, frequently without reference to the other field. There is a possibility that researchers in one domain will spend time researching problems that have been already solved in the other. Good science should involve a multi-perspective approach to problem solving so that research effort is not wasted on problems that have already been solved. Given this, it is imperative that researchers from both fields consider problems in a broader context, at the very least start to compare notes, if not consider active and intentional collaboration.**

### 1.1. Federated Learning

Federated Learning (FL) is the process of training a centralised *global* model with decentralised data (McMahan et al., 2016). A global model is initialised at the server and sent out to all of the clients. It is then trained on each client for a number of local epochs before the local weights are sent back to the central server for aggregation.

The archetypal aggregation method, Federated Averaging (FedAvg, (McMahan et al., 2016)) works by aggregating the local model weights by taking a weighted average, where

weightings are derived by the proportion of the global data that is on each client, such that a client with twice as much data will be weighted twice as much.

Once the model is aggregated at the server, it is then redistributed to the clients for more local training. This process is repeated for a number of rounds, or until a convergence threshold is reached.

FL frequently occurs in settings concerned with privacy preservation and/or where it is not practical to aggregate all of the data to one place for model training (e.g. training a federated language model for next word prediction in a virtual keyboard for smartphones (Hard et al., 2018)).

If the distribution of the global data across clients is Independent and Identically Distributed (IID), FedAvg and similar aggregation strategies converge to well performing models. However, the IID setting is highly restrictive and not realistic in real word applications. In non-IID situations, some local client models tend to drift away from the global model. This can result in poor convergence as the aggregation process can cause drifting clients to cancel out and make the global model update very small. This is referred to as "client drift" (Shi et al., 2022). A large amount of FL literature concerns strategies for convergence in non-IID client data distributions (Sahu et al., 2018), (Ghosh et al., 2021), (Reisser et al., 2021). Figure 1 illustrates a typical example of FL.

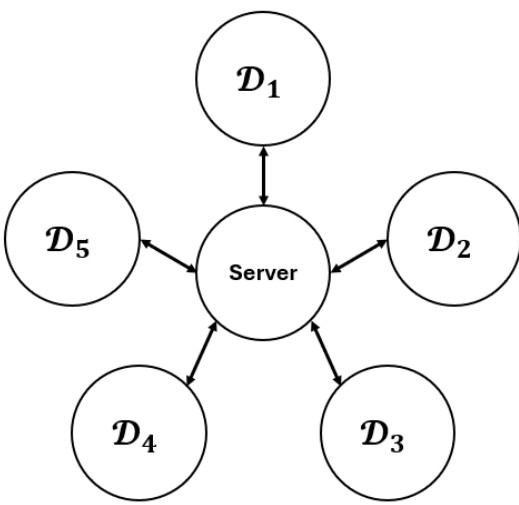

*Figure 1.* A standard 'star' network topology for centralised federated learning, with five clients, each with a partition of available data $\mathcal{D}_z$ and a central server. No data is communicated between clients and server.

### 1.2. Continual Learning

Continual Learning (CL) describes the task of machine learning when the data stream is changing (Wang et al.,

2024). A common scenario might be training a machine learning model for a task on data that is available today, retraining that model on data received tomorrow and then repeating this process over several days.

In an ideal scenario, the underlying distribution of the data does not change each day and therefore the model remains robust and well fitted to its training data. However, this ideal is unrealistic for common applications of CL, such as fraud detection, which is difficult due to concept drift (e.g. customers' habits evolve) (Dal Pozzolo et al., 2015). Concept drift is when the relation between the available data and the target variable changes over time (Lu et al., 2020) (Gama et al., 2014). This phenomenon was first observed by Ratcliff (1990) and McCloskey & Cohen (1989). In order to combat this, it is necessary to retrain the model as new data comes in. However, training on new data can lead to a model "forgetting" its previous capabilities. This is referred to as "Catastrophic Forgetting". Figure 2 illustrates a typical CL example.

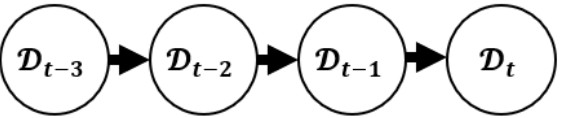

*Figure 2.* A standard CL setup where the partition of data available to a model at time $t$ varies. The arrows indicate that the model sees only data in the present (and the past in the case of replay buffers). In some sense the model in the future can be considered 'private'.

### 1.3. Federated Continual Learning

Federated Continual Learning (FCL) refers to a form of learning wherein data is partitioned both spatially and temporally (Criado et al., 2022). The training difficulties from both FL and CL apply, but they can each occur across two different axes. Data across federated clients can be non-IID making it different for a model at a given time step to convergence. Additionally, concept drift might occur as the distributions of the available data on each client changes over time. Figure 3 illustrates the added complexity of this problem.

## 2. A Unified Formalism of Partitioned Learning

In this section, we propose a unified theoretical framework, under the umbrella term "Partitioned Learning", that views CL and FL as equivalent instances of a single optimisation problem: *Risk Minimisation over Partitioned Distributions*. By abstracting the index of the data partition, whether it represents a temporal step or a spatial location, we can rigor-

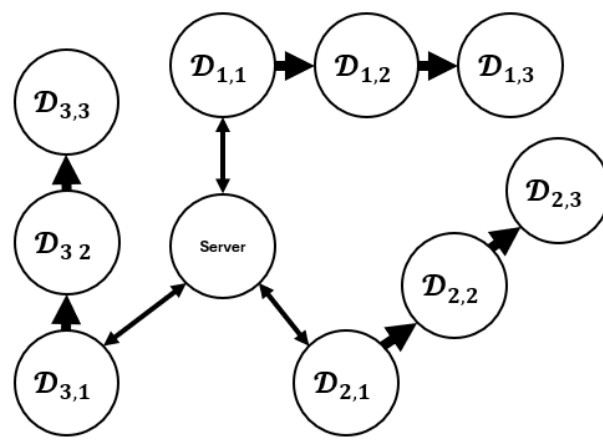

*Figure 3.* A decentralised federated learning setup where each of the client data partitioners are themselves continual. This paradigm is called Federated Continual Learning.

ously define the core hypotheses and assumptions regarding the global optimum as a prior, and classify the failure modes under common mathematical definitions. This underpins our entire position: because we can describe failure modes in this formalism that is consistent between both fields, it is clear that work considering these failure modes should be a combined effort.

## 2.1. Problem Formulation

Let $\mathcal{X} \subseteq \mathbb{R}^d$ be the input space and $\mathcal{Y}$ be the output space. We define a global data distribution $\mathcal{D}$ over $\mathcal{X} \times \mathcal{Y}$. We assume that $\mathcal{D}$ is not directly measurable as a monolith, but is instead composed of a mixture of $Z$ distinct component distributions (or contexts), $\{\mathcal{D}_z\}_{z=1}^Z$.

**Definition 2.1** (Partitioned Global Distribution). The global distribution $\mathcal{D}$ is defined as a mixture model:

$$\mathcal{D}(x, y) = \sum_{z=1}^{Z} \alpha_z \mathcal{D}_z(x, y) \qquad (1)$$

where $\alpha_z \in [0, 1]$ represents the weight (or prevalence) of partition $z$, such that $\sum_z \alpha_z = 1$.

In our Partitioned Learning framework, the index $z$ serves as an abstraction for the dimension of separation:

- **Temporal Axis (CL):** $z$ denotes a time-step $t$, where partitions are accessed sequentially.

- **Spatial Axis (FL):** $z$ denotes a client index $k$, where partitions are accessed in parallel but in isolation.

## 2.2. The Optimisation Objectives

We consider a Partitioned Learning setting with a model parametrised by $\theta \in \Theta$. We define the expected risk (loss)

of the model with respect to a specific partition $z$ as:

$$\mathcal{L}_z(\theta) := \mathbb{E}_{(x,y)\sim\mathcal{D}_z}[\ell(f_\theta(x), y)], \qquad (2)$$

where $\ell$ is a sample-wise loss function. Consequently, the global objective is to minimise the global risk:

$$\mathcal{L}_{\text{global}}(\theta) := \mathbb{E}_{(x,y)\sim\mathcal{D}}[\ell(f_\theta(x), y)] = \sum_{z=1}^{Z} \alpha_z \mathcal{L}_z(\theta). \quad (3)$$

We define two classes of optimal parameters which are central to our argument:

**Definition 2.2** (Specialist vs. Generalist Optima). The two classes of optimal parameters in Partitioned Learning as defined as:

1. The **Specialist Optimum** $\theta_z^*$ is the minimiser of the risk for a specific partition $z$:

$$\theta_z^* \in \arg\min_{\theta\in\Theta} \mathcal{L}_z(\theta). \qquad (4)$$

2. The **Generalist (Global) Optimum** $\theta^*$ is the minimiser of the global risk:

$$\theta^* \in \arg\min_{\theta\in\Theta} \mathcal{L}_{\text{global}}(\theta). \qquad (5)$$

## 2.3. Core Hypotheses: The Model as a Prior

The fundamental justification for pursuing a single model $\theta^*$ rather than an ensemble of specialists $\{\theta^*\}_{z=1}^Z$ relies on the assumption that the partitions share underlying structural commonalities. We formalise this via two hypotheses.

**Assumption 2.3** (Existence of a Shared Solution). We assume the existence of a parameter configuration $\theta^*$ within the hypothesis space that achieves bounded loss across all partitions. That is, for some small constant $\epsilon > 0$:

$$\forall z \in \{1, \dots, Z\}, \quad \mathcal{L}_z(\theta^*) \le \epsilon. \qquad (6)$$

*Hypothesis* 1 (The Transfer Hypothesis). We hypothesise that the global optimum $\theta^*$ serves as a superior initialisation (prior) for any specific partition $z$ compared to an uninitialised (random) model $\theta_{\text{init}}$. Formally:

$$\mathcal{L}_z(\theta^*) \ll \mathcal{L}_z(\theta_{\text{init}}). \qquad (7)$$

*Remark* 2.4. Hypothesis 1 is the driving force behind both CL and FL. It implies that information learned from $\mathcal{D}_{\neg z}$ (data outside partition $z$) contains transferable features that reduce the sample complexity of learning $\mathcal{D}_z$. In CL, this justifies maintaining knowledge from past tasks; in FL, it justifies aggregating gradients from remote clients.

## 2.4. Taxonomy of Failure Modes

The central challenge in learning $\boldsymbol{\theta}^*$ arises because the optimiser typically performs "greedy" updates on a single partition (or a subset) $\mathcal{D}_z$ at any given step, rather than optimising $\mathcal{L}_{\text{global}}$ directly. We define two distinct failure modes that result from this limitation: one dynamic and one static.

### 2.4.1. FAILURE TYPE I: DYNAMIC DIVERGENCE (FORGETTING AND DRIFT)

This failure describes the trajectory of the optimisation. Let $\boldsymbol{\theta}_{\text{old}}$ be the model parameters prior to an update step on partition $z$, and $\boldsymbol{\theta}_{\text{new}}$ be the parameters after optimising on $\mathcal{L}_z$.

We define *Dynamic Divergence* as the event where the local optimisation improves local performance but degrades global performance:

**Definition 2.5** (Dynamic Divergence). A learning update on partition $z$ exhibits Dynamic Divergence if:

$$\underbrace{\mathcal{L}_z(\boldsymbol{\theta}_{\text{new}}) < \mathcal{L}_z(\boldsymbol{\theta}_{\text{old}})}_{\text{Local Improvement}} \quad \text{AND} \quad \underbrace{\mathcal{L}_{\text{global}}(\boldsymbol{\theta}_{\text{new}}) > \mathcal{L}_{\text{global}}(\boldsymbol{\theta}_{\text{old}})}_{\text{Global Degradation}}.$$
(8)

An example of this can be seen in Figure 4, where the dashed purple optimisation trajectory diverges away from viable values of the old optimum in favour of the new optimum. Meanwhile the black trajectory moves towards a new optimum whilst maintaining performance for the old optimum, due to some kind of regularisation.

This formalism captures the core phenomenon in both domains:

- In **Continual Learning**, this is **Catastrophic Forgetting**: The model adapts to the current task $t$ but incurs a large increase in loss on tasks $1, \ldots, t-1$.

- In **Federated Learning**, this is **Client Drift**: The local update moves towards $\boldsymbol{\theta}_z^*$ (the client's local minimum), diverging from the server's consensus path towards $\boldsymbol{\theta}^*$.

### 2.4.2. FAILURE TYPE II: CONCEPT INCOMPATIBILITY (TASK CONFLICT)

This failure describes the geometry of the loss landscape, specifically regarding the relationship between the Specialist and Generalist optima. It represents a scenario where the tasks are inherently conflicting.

**Definition 2.6** (Concept Incompatibility Gap). For a given partition $z$, the Concept Incompatibility Gap $\Delta_z$ is defined as the performance deficit of the best possible global model compared to the best possible specialist model:

$$\Delta_z := \mathcal{L}_z(\boldsymbol{\theta}^*) - \mathcal{L}_z(\boldsymbol{\theta}_z^*).$$
(9)

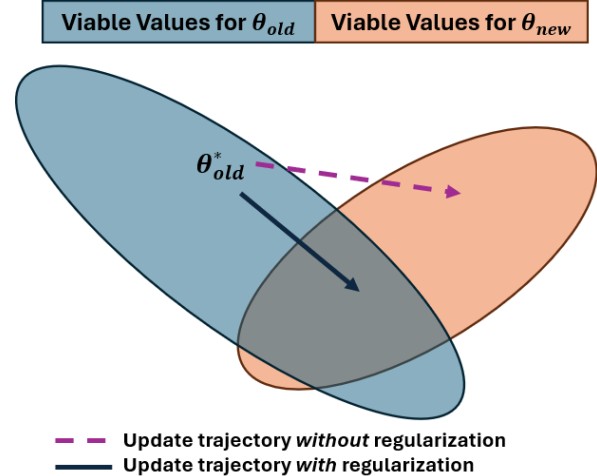

*Figure 4.* Dynamic Divergence is where an update of the model from its prior $\boldsymbol{\theta}_{\text{old}}$ towards $\boldsymbol{\theta}_{\text{new}}$ leads the model to forget/drift from its prior knowledge. In CL, $\boldsymbol{\theta}_{\text{old}} = \boldsymbol{\theta}_{t-1}$ (temporal prior); in FL, $\boldsymbol{\theta}_{\text{old}} = \boldsymbol{\theta}_{\text{global}}$ (spatial prior). In FCL, Dynamic Divergence can happen along either axes (or both).

An illustration of task conflict can be found in Figure 5, where there is no region in parameter space that sufficiently minimises both respective losses.

If $\Delta_z > 0$, it implies that $\boldsymbol{\theta}^*$ lies on the Pareto frontier of a multi-objective optimisation problem, requiring a trade-off between global and local solution parameters. Note that Dynamic Divergence can arise even when $\Delta_z = 0$, however, non-zero $\Delta_z$ *implies* Dynamic Divergence under both CL and FL learning paradigms.

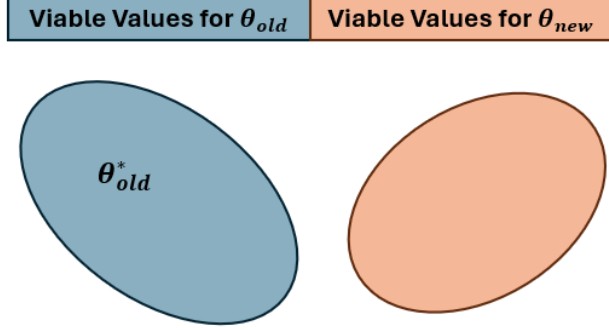

*Figure 5.* A concept incompatibility gap $\Delta_z > 0$, wherein the underlying assumption that a shared solution exists for every partition is not met, occurs in the event that the tasks in the data are inherently conflicting. In FCL task conflict can occur for spatial partitions, temporal partitions or both!

## 3. Correspondence between global data heterogeneity and partitioned model failure modes

Equipped with a joint formalism for Partitioned Learning and a taxonomy of failure modes, we will now consider the correspondence between global data heterogeneity (i.e. concept drift) and the failure modes that occur. We will see that this correspondence is setting agnostic within Partitioned Learning. This means that any results on detecting types of drift will be valuable for both FL and CL researchers.

Recall our definition of the global distribution $\mathcal{D}$ as a mixture of partitions $\mathcal{D}_z$. We recover the IID setting specifically when the mixture is homogeneous - when for any two partitions $z$ and $z'$, the joint distributions are identical: $\mathcal{D}_z(x, y) = \mathcal{D}_{z'}(x, y)$.

This expression can be written in terms of conditional and marginal distributions of $\mathcal{D}_z$ using Bayes' Theorem (Joyce, 2021):

$$\mathcal{D}_z(x, y) = \mathcal{D}_z(x)\mathcal{D}_z(y|x). \qquad (10)$$

This decomposes the IID condition into two separate conditions:

1. Marginal feature distributions are identical between mixtures: $\mathcal{D}_z(x) = \mathcal{D}_{z'}(x)$

2. Label distributions conditional on features are identical: $\mathcal{D}_z(y|x) = \mathcal{D}_{z'}(y|x)$.

These two conditions both being true is equivalent to the global data distribution being equivalent to the distribution of any and all individual of the mixtures: $\mathcal{D}(x, y) = \mathcal{D}_z(x, y) \ \forall (x, y)$.

The nomenclature between FL and CL for how IID data constraints can not be met varies but the underlying mechanisms are the same. We consider the domain task of training autonomous cars using collected image data as in previous work (Criado et al., 2022).

### 3.1. Case 1: Virtual Concept Drift - Invariant Posterior, Divergent Prior

In this instance, clients have samples from different domains but share the same targets:

$$\mathcal{D}_z(x) \neq \mathcal{D}_{z'}(x) \ and \ \mathcal{D}_z(y|x) = \mathcal{D}_{z'}(y|x). \qquad (11)$$

- In the context of FL this might look like different clients having data from some cars that drive on the left and some that drive on the right.

- For CL, this could appear as getting new data from cars driving in new regions ("virtual concept drift").

Whilst the specialist optima $\theta_z^*$ might differ, the generalist optimum $\theta^*$ likely remains valid because the labelling function hasn't changed. If the feature space is different between partitions but there is no conflicting labels then our concept incompatibility gap should theoretically be zero: $\Delta_z = 0$.

However, the optimiser sees a biased sample space: when training on a partition $z$, the gradients are calculated only over the support of $\mathcal{D}_z$. This pulls the weights away from the support of another partition $\mathcal{D}_{z'}$. This a clear example of Failure Type 1: Dynamic Divergence. The key to addressing this is to use regularization to force updates from different clients to stay within a mutual support of one another.

### 3.2. Case 2: Real Concept Drift - Divergent Posterior, Invariant Prior

In this case, the feature spaces are analogous but the labels differ:

$$\mathcal{D}_z(x) = \mathcal{D}_{z'}(x) \ and \ \mathcal{D}_z(y|x) \neq \mathcal{D}_{z'}(y|x). \qquad (12)$$

- In FL this may be exemplified by changing traffic laws by location, where the feature is a car approaching a yellow light, but the label is the appropriate reaction to the yellow light (stop/go).

- In CL this would be the same example but considering changing traffic laws across time. This type of concept drift is referred to as real concept drift.

In this scenario, the concept incompatibility gap will be strictly positive $\Delta_z > 0$, meaning a single global model $\theta^*$ cannot perfectly solve all partitions simultaneously. This is a clear example of Failure Type II: Concept Incompatibility. The key to addressing this is to identify incompatible data sources and separate them into different systems.

### 3.3. Case 3: Total Concept Drift - Divergent Posterior and Divergent Prior

There is also a case where there can be both different features and different labels.

$$\mathcal{D}_z(x) \neq \mathcal{D}_{z'}(x) \ and \ \mathcal{D}_z(y|x) \neq \mathcal{D}_{z'}(y|x). \qquad (13)$$

This is referred to as *Total Concept Drift* in CL literature (a mixture of Virtual and Real concept drift) and is when both failure types occur at once: participants want to learn a common task (driving), but their input spaces are significantly unequal, and their reactions to some of the inputs are different too. Here, both regularization and separation of incompatible data sources are required!

The magnitude of these distributional shifts (skewness) directly correlates to the risk of Dynamic Divergence defined

in Section 2.4.1. The greater the distance between $\mathcal{D}_z$ and $\mathcal{D}_{z'}$, the more the greedy update $\theta_{new}$ on partition $z$ will degrade the global loss $\mathcal{L}_{global}$.

# 4. Strategies to address issues across Partitioned Learning take direct inspiration from one another

There are a range of approaches in the CL and FL literature to approach virtual, real and total concept drift. Whilst some strategies have emerged from the CL and FL literature organically and independently, we note a clear value for comparing notes across fields. This is especially, but not exclusively, true in the case of FCL approaches. We look at current advances in CL literature, namely Behrouz et al. (2025) and Darlow et al. (2025), and argue how their distributed flavour shows some surprising symmetry with FL systems. This is particularly true for distributed systems where updates are not perfectly synchronised. We finish by outlining how advances to approximate the Concept Incompatibility Gap can be treated broadly as a Partitioned Learning problem (rather than specifically an FL or CL problem). This highlights that comparing notes is necessary and collaboration would be mutually beneficial for researchers across FL and CL.

## 4.1. Approximating the Global Distribution (Replay Buffers)

One strategy to combat Dynamic Divergence is to share data across partitions. In CL literature this is referred to as using *replay buffers* (Rolnick et al., 2018). This involves storing a few examples from previous data partitions, allowing updates for a new training round to be based on both old and new training data. In FL literature this is referred to having a *public transfer set* or a *dataset for distillation* (Li & Wang, 2019). It involves distributing a small public dataset across all federated clients.

The core idea in both is to change the partition specific loss functions $\mathcal{L}_z$ so that they better approximate the global loss $\mathcal{L}_{global}$. By mixing old/global data into the current local update, the gradient $\nabla \mathcal{L}_{local}$ is less likely to lead to a solution that increases $\mathcal{L}_{global}(\theta_{new})$.

For both CL and FL, deciding what data to use is a challenge. In CL, assuming all data is equally easy to access, randomly selecting data can lead to poor sampling across the distribution of features and labels. To combat this, cluster centroids of the features of incoming data can be used as a representative data pool for the replay buffer (Solomon et al., 2025).

Privacy preservation is often the motivation for using a federated system (Cheng et al., 2020), meaning the availability

of data for FL is typically less generous.

Machine Learning models tend to perform best on the majority classes in data (e.g. a skin cancer classifier will likely perform better on the skin tones it sees more of during training (Pope et al., 2025)). FL aggregation methods tend to exacerbate this as they generally take weighted averages of client models (McMahan et al., 2016; Sahu et al., 2018).

Partitioned Learning systems will perform better on the public data they have, because all clients update with respect to it (Li & Wang, 2019). In order for a Partitioned Learning model to perform well for its minority classes, the public data should accurately represent those minority classes. Historically, groups of people under-represented in data are also groups that have been subjugated by data collected on them. This is called the "Paradox of Exposure" - where those who stand to significantly gain from being counted are in the most danger from that same counting (or classifying) act (D'Ignazio & Klein, 2020). Therefore, deciding on which data items to make publicly available can be difficult due to attempts to balance personalisation for minority classes with the Paradox of Exposure.

There are important social and political considerations that are pertinent when using replay buffers in Partitioned Learning systems and research into these considerations will be relevant to both (Doh et al., 2025).

## 4.2. Constraining the Update (Regularization)

Under our unified view, both FL and CL algorithms can be viewed as methods to approximate $\boldsymbol{\theta}^*$ while mitigating Dynamic Divergence. This can be achieved by introducing a regularization term that constrains the greedy update on $\mathcal{D}_z$ to remain proximal to a *prior* derived from the other partitions:

$$\boldsymbol{\theta}_{\text{new}} \leftarrow \arg\min_{\boldsymbol{\theta}} \left( \mathcal{L}_z(\boldsymbol{\theta}) + \lambda \Omega(\boldsymbol{\theta}, \boldsymbol{\theta}_{\text{prior}}) \right). \quad (14)$$

In CL, $\boldsymbol{\theta}_{\text{prior}} = \boldsymbol{\theta}_{t-1}$ (temporal prior); in FL, $\boldsymbol{\theta}_{\text{prior}} = \boldsymbol{\theta}_{\text{global}}$ (spatial prior).

A common approach to mitigate the effects of Catastrophic Forgetting in CL is Elastic Weight Consolidation (EWC) (Kirkpatrick et al., 2017). It was inspired by the neuroscientific concept of *synaptic consolidation*. EWC uses approximations of the diagonal elements of the Fisher Information Matrix, which is a method of measuring the amount of information an observable random variable has, within the model's parameters. In terms of (14) EWC has regularization term:

$$\Omega_{EWC}(\boldsymbol{\theta}_t, \boldsymbol{\theta}_{\text{t-1}}) = \sum_i \frac{1}{2} F_i (\theta_{t,i} - \theta_{t-1,i}^*)^2. \quad (15)$$

In FL, a common aggregation strategy which works to gen-

eralize FedAVG to heterogeneous settings is FedProx (Sahu et al., 2018). Rather than optimise for the client loss directly $\mathcal{L}_z(\theta)$ as in FedAvg, FedProx adds a proximal term that leashes the client updates to the model updates. In terms of (14) FedProx has regularization term:

$$\Omega_{FedProx}(\boldsymbol{\theta}, \boldsymbol{\theta}_{\text{global}}) = \frac{1}{2}||\theta - \theta^*_{global}||^2. \quad (16)$$

The penalty in FedProx is isotropic (L2 norm of the difference between the new weights), effectively assuming all weights are equally important. To account for this, researchers have taken inspiration from EWC and integrated a non-isotropic penalty term in the form of FedCurv (Shoham et al., 2019). This solution is not without its drawbacks: the authors' note that their method involves an increased computational requirement for the clients to calculate their relevant Fisher Information (Shoham et al., 2019).

These methods assume a good Generalist Optimum ($\theta^*$) exists ($\Delta_z \approx 0$). Their goal is purely to constrain the optimiser's trajectory ($\theta_{\text{new}}$) to prevent greedy local updates from moving too far away from the prior ($\theta_{\text{init}}$), thus mitigating forgetting/client drift - see Figure 4.

### 4.3. Variable Update Rates (Biologically Inspired CL approaches)

In CL, recent developments including Nested Learning (Behrouz et al., 2025) and Continuous Thought Machines (Darlow et al., 2025) seek to emulate biological learning mechanisms by varying the rates for updates across networks.

Consider a a Convolutional Neural Network (CNN) trained to recognise cars in a CL setting, which is shown images of different kinds of cars over the years. A CNN trained on this task might learn early layers that typically capture basic features like vertical lines and curves, while later layers synthesize these into complex objects such as cars. When training a CNN to recognize vehicles over several years in a Continual Learning setting, these fundamental early-layer abstractions remain largely constant and *require* few updates. Conversely, because car designs and styles evolve, the later layers must update more frequently to adapt to these shifting visual patterns.

Architectures that decompose deep learning models into distributed components, each with unique parameters and varying update frequencies, represent a promising frontier for Continual Learning (Behrouz et al., 2025). A primary challenge in this approach is managing multiple sub-models while maintaining a unified global objective. This difficulty is precisely mirrored in Federated Learning, particularly within large-scale Internet of Things (IoT) networks, where partitioned models frequently communicate at inconsistent

intervals (Nguyen et al., 2021). Established FL aggregation strategies, such as FedProx, were specifically engineered to remain robust despite these desynchronised client updates(Sahu et al., 2018). Consequently, CL researchers exploring variable update rates can accelerate their progress by adopting these proven FL strategies for handling asynchronous data streams.

### 4.4. Partitioning the Parameter Space (Clustering and MoE)

The approaches discussed above addressed Dynamic Divergence; this section will consider the presence and solutions to concept incompatibility. Here the common theme is 'divide and conquer', with varying approaches in what to divide by and how best to conquer.

In CL, models experience this negative transfer when learning on conflicting data distributions. Mixture of Experts (MoE) architectures (Jacobs et al., 1991) have been implemented, which involve a set of "expert" models and a gating network/router model deciding which model should be used for each data point. As CL typically does not have privacy constraints, the data distribution can be directly examined in an online setting (Aljundi et al., 2016).

In FL, we cannot directly examine data centrally due to privacy constraints. Instead, existing work looks to cluster devices together, either after a stationary loss has been reached (Sattler et al., 2019) or as part of the aggregation strategy (Ghosh et al., 2021), to create sub-federated systems. This works well under the assumption that an individual data partition does not have conflicting internal tasks (which is not always a safe assumption, particularly in FCL).

MoE has been applied in FL paradigms, as in the case of Federated Mixture of Experts (FedMix) (Reisser et al., 2021), where the clients are data sources and the experts are specialised global models. The client computes gating weights locally and sends aggregated gradients for the experts, since the server cannot see data from individual clients. Recent work has used MoE in a FCL setting to adapt to concept drift across federated clients (Bhope et al., 2025).

All of these methods directly address the structural problem where a single $\theta^*$ is insufficient (i.e., when the Concept Incompatibility Gap $\Delta_z > 0$). They achieve this by abandoning the single global model assumption entirely. Instead, they learn a set of specialists $\{\theta_1, \ldots, \theta_K\}$, effectively converting the problem into $K$ smaller, less incompatible sub-problems. The resulting smaller problems (e.g., within one cluster of clients) still face divergence, but to a lesser degree.

### 4.4.1. CALCULATING THE CONCEPT INCOMPATIBILITY GAP

In current practice (Li et al., 2025a;b), the Concept Incompatibility Gap is rarely measured directly; instead, it is implicitly addressed during training. If a generalist model $\theta^*$ fails to reach acceptable performance despite sufficient capacity, we assume a high gap exists and pivot to "divide and conquer" strategies. While we can use the current model's performance to approximate this gap, these empirical proxies are inherently limited by the model's architecture and the quality of the function approximation.

Although we can approximate the gap, a rigorous analytical method for determining it remains elusive. This work demonstrates that the root of task conflict across both Federated and Continual Learning is *Real Concept Drift*, where the input-output mapping itself diverges. At this point, the utility of a "shared prior" reaches a limit: the prior may still provide useful features, but the specific weights must be significantly altered to avoid massive error. Moving beyond "trial and error" to a deeper theoretical understanding of this incompatibility is a major opportunity that benefits both FL and CL research communities.

## 5. Alternate Views

A reasonable objection to our unified framework is that the operational constraints and objectives of FL and CL are fundamentally incompatible in the real world. One could argue that the core restriction in FL is privacy preservation, whereas in CL it is memory and adaptation, and a unified formalism of their tasks descriptions is not practically useful. For example, widely used CL strategies like replay buffers rely on storing historical data, which typically violates the core privacy requirements of FL clients. Additionally, one could argue that the objectives diverge. In FL we seek a consensus model ($\theta^*$) that aims to model all clients' data simultaneously. Meanwhile in CL we prioritise temporal adaptation, where "forgetting" outdated concepts is often a necessary feature, not a bug, of tracking non-stationary data.

While we concede that a direct algorithmic copy-and-paste is likely to be insufficient, we argue this engineering mismatch does not negate practical benefits of the unified formalism. The core mathematical challenge remains identical: approximating a diverse 'global' distribution $\mathcal{D}$ using only locally relevant (in time/space), partitioned gradients $\nabla \mathcal{L}_z$.

The aforementioned constraints dictate the implementation, but both fields look to approximate $\mathcal{D}$ and then allow for plasticity (CL) or personalisation (FL) when virtual concept drift occurs or divide and conquer strategies when task conflict occurs. Both fields independently recognise the same mathematical challenge, and thus we conclude that:

**treating CL and FL separately obscures their shared mathematical structure and precludes cross-pollination of ideas that drive algorithmic developments in both fields.** Our position embraces the practical incompatibilities of each field, proposes a unified perspective on their mathematical objectives, and pushes towards more referencing between CL and FL research.

## 6. Conclusion

Federated Learning and Continual Learning can both be described as Partitioned Learning problems - problems where you seek to minimise risk over partitioned distributions of data. In this work we have formalised this link and shown that the failure modes and their causes in terms of the type of partition data heterogeneity are equivalent. Given this formalism, we can now see that the recency bias seen in Continual Learning is comparable to a client side bias in Federated Learning.

Research into strategies for mitigating model failure have naturally advanced without exchange of ideas in some instances such as replay buffers/using a public dataset. However, we have demonstrated that advances have been made when researchers have compared notes and considered inspiration from other literature (in the example of FedCurv (Shoham et al., 2019)). We have shown that current research in Continual Learning is considering more distributed learning systems with varying update weights (Darlow et al., 2025; Behrouz et al., 2025). We then argued that managing these types of distributed learning systems is an area extensively explored in Federated Learning literature. We have also argued that an open area for both fields to explore is developing a rich mathematical understanding of the Concept Incompatibility Gap and deducing when this type of task conflict occurs beyond a simple trial and error approach.

If more Federated Learning and Continual Learning researchers compared notes, they would be able to benefit from the exchange of ideas between disciplines that look at Partitioned Learning problems from different perspectives. This would reduce the amount of repetition in research, allow both sides to take inspiration from one another, and motivate the pursuit of problems with broader concern that previously thought.

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
