# OpenReview forum: "Position: Federated Learning and Continual Learning researchers should compare notes"
_ICML.cc/2026/Position_Paper_Track — Submitted to ICML 2026 Position Paper Track_

### Official Review · Reviewer_Ki1r · 2026-03-11

**Significance:** 2
**Argument Clarity:** 3
**Rating:** 3
**Confidence:** 4

**Questions:**

The temporal-spatial asymmetry seems to fundamentally limit the algorithmic transfer between the two areas. Could you clarify which specific algorithms or techniques you believe transfer smoothly in each direction?

**Alternative Views Section:**

Yes

**Compliance With Llm Reviewing Policy A Conservative:**

Affirmed.

**Discussion Potential:**

2

**Final Justification:**

While there exists high-level mathematical symmetry in certain aspects between FL and CL, the asymmetric nature of algorithmic knowledge transfer between the two fields remains inadequately analyzed and discussed, which makes the central position insufficiently motivated from my perspective. Therefore, I maintain my score.

**Paper Summary:**

The paper argues that researchers should compare notes from Federated Learning (FL) and Continual Learning (CL), as they can both be considered as optimization over partitioned distributions, so that research effort is not wasted on problems that have already been solved. The papers provide a unified formalism of partitioned learning and define shared failure modes (dynamic divergence and concept incompatibility). The authors then make connections between global data heterogeneity and these modes, and discuss existing approaches to address these issues arising from partitioned learning, where similarities of the approaches used in FL and CL have been illustrated. Finally, the authors provide an alternative view regarding the incompatible operational constraints and objectives of FL and CL.

**Position:**

Yes

**Position In Title:**

Yes

**Related Work:**

2

**Strengths And Weaknesses:**

Strengths:

1. The writing is easy to follow. The intuition of the connections between FL and CL is clear.

2. The introduction of three cases of concept drift is interesting.

Weaknesses:

1. The relation and discussion to multi-task learning (MTL) are inadequate and missing. The problem formulation, the concept incompatibility, and the approaches to solve task conflict, etc, are highly related to MTL, which has a rich literature and should be discussed properly.

2. The (temporal and spatial) asymmetry between FL and CL is not adequately resolved and discussed in the paper. In FL, the information (gradients, function values, etc,.) about each local dataset can be computed independently and be aggregated simultaneously,  while this is not the case for CL. This fundamental difference significantly distinguishes the optimization methods in FL and CL. This has not been properly discussed and motivated in the paper, which weakens the reasoning of the proposed position.

3. The call to action is too general and a bit vague. The position says we should compare notes between these two areas of research. However, there is no concrete action about which open problems and which comparisons we should investigate.

Minors:
The paper does not provide concrete or real-world examples of the proposed notions. For instance, for which problems do we have dynamic divergence, concept incompatibility, and various concept drifts for both FL and CL tasks?

**Support:**

2

---

> ### Author Rebuttal · Authors · 2026-03-30
>
> We sincerely thank the reviewer for identifying these vital connections, particularly regarding Multi-Task Learning (MTL) and the practical asymmetries between the fields.
>
> Regarding Multi-Task Learning (MTL): We agree that MTL is deeply connected to this framework. We will add a dedicated discussion in the revision framing MTL as the idealised, unconstrained baseline of Partitioned Learning. In MTL, there are no "spatial or temporal walls," allowing for simultaneous access to all data/gradients. The Concept Incompatibility Gap ($\Delta_z$) we define is essentially the measure of negative transfer in MTL.
>
> Regarding the asymmetry between FL and CL, we thank you for pointing out the fundamental asymmetry in implementation. We will expand our discussion to explicitly address this. Specifically, we will highlight the theoretical symmetry: a CL model with infinite "stability" (no forgetting) converges to the global FL consensus model, while a CL model with infinite "plasticity" converges to the isolated local FL client models. The performance delta between these states is exactly the Concept Incompatibility Gap.
>
> The utility of the partitioned learning perspective has been informally recognised by CL researchers at the gradient aggregation level. For example, to overcome the sequential restriction, CL methods like Gradient Episodic Memory (GEM) (https://arxiv.org/abs/1706.08840) explicitly store and project past task gradients, essentially simulating a parallel FL gradient aggregation step locally over time. We will add these specific examples to strengthen the reasoning for CL and FL researchers to leverage our theoretical formalisation.
>
> Regarding the Call to Action, as detailed in our response to Reviewer 1, we are adding a dedicated section proposing specific heuristic steps, cross-paradigm benchmarks, and algorithmic cross-pollination (e.g., applying CL network-embedding stability guarantees via SVD directly to privacy-preserving Federated SVD systems). We will also include the gradient-surgery adaptations mentioned above (MTL/CL).
>
> Regarding Real-World Examples: We provide real-world examples in Sections 3.1, 3.2, and 3.3 (such as  of autonomous driving). For instance, cars driving in different regions (left vs. right side of the road) represent Virtual Concept Drift , while changing traffic light laws represents Real Concept Drift. We will revise the text to ensure these examples are explicitly and visually mapped to Dynamic Divergence (Failure Type I) and Concept Incompatibility (Failure Type II).

---

> > ### Author Rebuttal · Reviewer_Ki1r · 2026-04-01
> >
> > I appreciate the rebuttal from the authors. On the discussion of MTL, I also encourage the authors to go beyond conceptual connections and discuss more specific MTL algorithms that address task conflict and how they relate to methods used in CL and FL.
> >
> > My main concern is still with the part of the FL-CL asymmetry. While there exists theoretical equivalence on the solution inside,
> > there are still fundamental differences in core development principles. FL can aggregate gradients simultaneously, which is a powerful operation without the need for asynchronous aggregation like GEM, while CL cannot. Moreover, the priority metrics used in FL (typically the communication cost) are also different from CL. These differences make the algorithmic design and analysis highly different, meaning that the direction and scope of useful knowledge transfer between the two fields can be asymmetric and context-dependent. Therefore, the central position remains a bit insufficiently motivated to me and
> > I decided to maintain my current score.

---

> > > ### Author Response · Authors · 2026-04-02
> > >
> > > Our strong position is that implementation aside, there has been a significant amount of redundant research in these fields (for example FedCurl and Elastic Weight Consolidation) and the reason for it is there are problems that are exactly the same between them, because on a high-level they are instances of partitioned learning that cover different axes.
> > >
> > > We acknowledge that sometimes the direction and scope of useful knowledge transfer is sometimes asymmetrical: CL has spent more time focused primarily on underlying concept drift, meanwhile FL is a newer field and has had split priorities including drift, privacy, asynchronous updates, as the reviewer points out! A CL researcher interested in distributed systems with potentially asynchronous updates (such as nested learning or CTMs) or a CL researcher interested in data being private between time updates could benefit from reading FL literature on this topic. Meanwhile FL researchers are always worried about non-IID setups so could arguably more consistently benefit from reading CL literature.
> > >
> > > Our current observations are that researchers in FL and CL respectively are unlikely to have encountered material from the other field. We hope with this work to convince researchers from both work that there are benefits to comparing notes due to the symmetries at a high level, even if specific nuance of implementation sometimes varies.

---

### Official Review · Reviewer_WXvS · 2026-03-12

**Significance:** 2
**Argument Clarity:** 3
**Rating:** 3
**Confidence:** 4

**Questions:**

1. Can you characterize conditions under which the Concept Incompatibility Gap is provably zero or strictly positive?
2. Does the unified view lead to any new algorithmic design principles beyond reinterpretation of existing methods?
3. Could you provide even a minimal experimental case study illustrating the practical benefit of treating CL as FL (or vice versa)?

**Alternative Views Section:**

Yes

**Compliance With Llm Reviewing Policy A Conservative:**

Affirmed.

**Discussion Potential:**

3

**Final Justification:**

Thanks for the response. Some of my concerns has been addressed. But there are still Weaknesses need to be solved and claimed.

**Paper Summary:**

This position paper argues that Federated Learning (FL) and Continual Learning (CL) can be understood within a unified framework termed Partitioned Learning. The authors formalize both settings as risk minimization over a mixture of partitioned distributions and introduce two shared failure modes: Dynamic Divergence (covering client drift and catastrophic forgetting) and Concept Incompatibility Gap (capturing task conflict). The paper’s main goal is to encourage stronger cross-referencing and collaboration between the FL and CL communities. Overall, the authors study the concept of interpreting FL and CL through a common optimization lens and consider a significant challenge in bridging two communities that often work on mathematically related problems without explicit coordination. The paper is clearly written and easy to follow, but I have concerns about the depth and novelty of the contribution.

**Position:**

Yes

**Position In Title:**

Yes

**Related Work:**

3

**Strengths And Weaknesses:**

(1) Strengths:
1. The paper reads smoothly and is logically structured. The figures are simple but effective in illustrating the geometric intuition.
2. The mixture formalism in Section 2 provides a clean way to express both spatial and temporal partitioning.
3. The discussion in Section 4 draws reasonable parallels between replay buffers vs. public datasets, EWC vs. FedProx/FedCurv, and MoE vs. clustered FL. For readers less familiar with one of the two areas, this cross-mapping is informative.
(2) Weaknesses:
1. Several existing papers have already discussed the relationship between federated learning, continual learning, and federated continual learning (e.g., Federated continual learning via knowledge fusion: A survey). Please clarify how your perspective differs from that presented in these relevant reviews.
2. While the formalism is clean, it does not substantially extend existing theory. The equivalence between distributed non-IID optimization and sequential non-stationary optimization has been informally recognized before. The paper sharpens the framing but does not introduce new theorems, bounds, or algorithmic insights.
3. Concept incompatibility gap is interesting, but the paper does not go further. There are no analytical results, no measurable proxy, and no guidance on how to estimate it in practice. Section 4.4.1 acknowledges this gap but leaves it entirely open. This makes the formalism feel incomplete.
4. Even for a position paper, some small synthetic example demonstrating the symmetry between CL and FL under the proposed framework would significantly strengthen the argument. At present, the claims remain theoretical and analogical.
5. Operational constraints differ in meaningful ways (privacy, communication limits, personalization objectives in FL; memory and adaptation priorities in CL). While the optimization objective may look similar, the engineering and deployment realities are not fully symmetric. This limits how far the unification can be taken.
6. The framework relies heavily on the existence of a meaningful generalist optimum. In many realistic settings (e.g., highly personalized FL or unlearning in CL), such a single optimum may not be desirable. This assumption deserves more scrutiny.

**Support:**

3

---

> ### Author Rebuttal · Authors · 2026-03-30
>
> We thank the reviewer for the thorough review and excellent questions.
>
> Regarding differences from FCL Surveys: Existing FCL surveys examine the intersection of the two fields, building systems that face spatial and temporal constraints simultaneously. Our position paper argues for the mathematical isomorphism between the two distinct fields. We are arguing that a researcher working purely on FL and one working purely on CL are solving the same geometric optimization problem, just along different axes. We will explicitly clarify this distinction in the text.
>
> Regarding the Concept Incompatibility Gap & Analytical Results: We highly value this critique. In response, we have formalized this gap with a theoretical result in the appendix. We proved that $\Delta_z$ is strictly bounded below by the KL divergence between the local and global posteriors: $\\mathbb{E}\_{x \\sim \\mathcal{D}(x)}[D\_{KL}(\\mathcal{D}\_z(y|x)||\\mathcal{D}\_{global}(y|x))]$. We also included a small experimental case study verifying this on MNIST to demonstrate how label flips strictly widen this gap. More details can be found in our rebuttal to reviewer 2.
>
> Regarding the Generalist Optimum: We agree that a single optimum is often undesirable (e.g., highly personalized FL). The formulation of $\Delta_z > 0$ is specifically designed to identify when a single optimum fails, serving as the mathematical trigger for 'divide and conquer' approaches like MoE or clustering.

---

> > ### Author Rebuttal · Reviewer_WXvS · 2026-04-03
> >
> > Thanks for the response. Some of my concerns has been addressed. But there are still Weaknesses need to be solved and claimed.
> >
> > Therefore, I keep my score.

---

### Official Review · Reviewer_wS4h · 2026-03-12

**Significance:** 2
**Argument Clarity:** 2
**Rating:** 2
**Confidence:** 3

**Questions:**

none

**Alternative Views Section:**

Yes

**Compliance With Llm Reviewing Policy A Conservative:**

Affirmed.

**Discussion Potential:**

1

**Paper Summary:**

The position advocated in this paper is that federated learning and continual learning are equivalent instances of a single optimisation problem, which is risk minimisation over partitioned distributions. For this purpose, the paper proposes a unified framework and mainly studies failure modes.

**Position:**

Yes

**Position In Title:**

Yes

**Related Work:**

2

**Strengths And Weaknesses:**

Strengths:
The stated position of the paper is sound. The paper provides a unified formalism through an optimization problem that bridges the gap between the federated learning and continual learning paradigms.

A major contribution of the paper is on describing failure modes in the proposed formalism and showing that these failure modes are consistent for both federated learning and continual learning. Moreover, the paper explores strategies to address underlying issues, by taking direct inspiration from one another.

The paper includes an Alternate Views section that clearly points out the major issues of the stated position.

Weaknesses:

Essentially, as pointed out in the Alternate Views section, federated learning and continual learning are intrinsically very different in practice. As the former seeks to address privacy preservation, the latter aims to address memory and adaptation issues. Thus, the resolution approaches are often antagonistic, such as storing historical data in continual learning to address catastrophic forgetting, which clearly violates the privacy-preservation of federated learning clients.

The reasoning of the stated position is sound from the theoretical aspects, namely proposing a unified framework. However, the proposed framework called risk minimisation over partitioned distributions seems a bit simplistic and not realistic, as discussed in the previous paragraph. Moreover, almost all machine learning problems are risk minimisation and there are many that consider partitioned distributions, such as the stochastic gradient descent and variance reduction for stochastic optimization.

As the proposed formalism seems to be synthetic and not much practical in the real world, we think that the paper is not significant enough and relevant to the ICML community, thus limiting its impact.

There are also some spelling and grammatical errors, which make some sentences difficult to follow, such as: “Data across federated clients can be non-IID making it different for a model at a given time step to convergence.” (is it “difficult” and not different ?)

**Support:**

3

---

> ### Author Rebuttal · Authors · 2026-03-30
>
> We appreciate the reviewer’s critical assessment.
>
> Regarding the Framework being Simplistic: We agree that a purely conceptual framework is insufficient. In response, we have added a theorem in the appendix, which establishes a rigorous information-theoretic lower bound on the Concept Incompatibility Gap. We prove that under Real Concept Drift, $\Delta_z$ is strictly bounded below by the expected KL divergence between the local conditional distribution and the global mixture distribution:
> $\\mathbb{E}\_{x \\sim \\mathcal{D}(x)}[D\_{KL}(\\mathcal{D}\_z(y|x)|| \\mathcal{D}\_{global}(y|x))]$.
> This theoretical result is true under idealised assumptions and demonstrates a practical way to measure the incompatibility gap. It doesn’t account for the noisiness of deep learning training, motivating further work to attempt to expand the theoretical results. We have also validated this result empirically on split-MNIST by progressively flipping some of the labels of the data, forcing task conflict. We will add all details of these experiments in the appendix of the camera ready version.
>
> Regarding Antagonistic Approaches (Privacy vs. Memory): We view these differing constraints as a feature, not a bug, of this comparison. Because CL has been able to operate in predominantly privacy free conditions, the bulk of the research in this area has been addressing concept drift of various forms. Meanwhile FL research has held privacy preservation as the main driving factor. This means that FL researchers can benefit from adapting CL methods to federated applications (such as taking inspiration from EWC to develop the FedCurve method). This also means that CL researchers faced with challenges familiar to FL researchers such as privacy constraints or reconciling distributed systems with asynchronous contributions can benefit from reading FL research.

---

> > ### Author Rebuttal · Reviewer_wS4h · 2026-04-04
> >
> > I thank the authors for the response. However, my major concerns remain unsolved.
> > I acknowledge that the authors will include experiments/appendix in a revision. However, I cannot assess future hypothetical modifications and results.
> > The same goes for taking inspiration from EWC to develop the FedCurve method, for example. This future working direction is also speculative. The paper would have benefitted from such a concrete example that demonstrate the interest of this work that aims to bridge the gap between two frameworks. I cannot assess that hypothetically this would be beneficial.

---

### Official Review · Reviewer_Vg97 · 2026-03-17

**Significance:** 3
**Argument Clarity:** 3
**Rating:** 3
**Confidence:** 4

**Questions:**

1. Figure 5 does not show the incompatibility gap anywhere. Is the figure incomplete?

**Alternative Views Section:**

Yes

**Compliance With Llm Reviewing Policy A Conservative:**

Affirmed.

**Discussion Potential:**

3

**Final Justification:**

The paper inadequately addresses fundamental differences between CL and FL. I believe the paper has several issues and is not good enough for acceptance.

**Paper Summary:**

The paper takes the position that Federated Learning (FL) with spatial constraints and Continual Learning (CL) with temporal constraints on model training are the same problem of “Partitioned Learning” along different axes. The authors argue that researchers working in these two fields should compare notes since they have similar strategies for mitigating failure modes. The paper propose a unified problem of risk minimization over partitioned distributions which includes both spatial and temporal partitions of data. The paper discuss different concept drift cases covering the two fields and strategies from one field which inspires the other field. The paper also discuss alternate views and finally urges researchers in the two fields to exchange ideas for mutual benefit.

**Position:**

Yes

**Position In Title:**

Yes

**Related Work:**

3

**Strengths And Weaknesses:**

Strengths -
1. The paper nicely points out the common themes in the two fields and discuss shared objectives and failure modes.
2. The formulation of Partitioned Learning covering both CL and FL is interesting.


Weaknesses-
1. There are some fundamental differences between the paradigms of federated and continual learning which has not been accounted for in the paper. This paper considers a subset of offline continual learning works which only deals with partitions of a given dataset incrementally appearing in subsequent tasks. I do not see discussions on more real-time settings like Online Continual Learning (OCL) or streaming continual learning which has been extensively studied over years. I find it hard to consider OCL as a partitioned learning problem. Another difference is - new data or tasks in CL is expected to keep coming over time while in FL we have finite number of clients (even though new clients can join).

2. While the paper provides many common themes between the two fields, the difference between the two fields is not just about temporal and space constraints, there are more significant differences. In CL, the same old model is updated on new tasks leading to the stability-plasticity dilemma whereas in FL every client keeps updating individual models by iterative aggregation. So, there are fundamental differences in the training dynamics, it is not just about the data partitions. While the paper briefly mentions this, I believe this is a significant difference.

3. The authors do not propose specific guidelines to merge the two fields or propose benchmarks and evaluation directions to treat the two fields as a partitioned learning problem. The paper does not provide a “Call to Action” section to outline or summarize the steps to realize the position.

**Support:**

2

---

> ### Author Rebuttal · Authors · 2026-03-30
>
> We sincerely thank the reviewer for the constructive feedback.
>
> Regarding OCL & Training Dynamics: We will clarify in the camera-ready version that Online Continual Learning (OCL) represents the extreme limit of Partitioned Learning where the partition size is minimal (e.g., $|\mathcal{D}_z| = 1$ or $\alpha_z \to 0$). While we agree the training dynamics differ in deployment (stability-plasticity vs. iterative aggregation), the geometric problem of a gradient update moving parameters away from a prior global/historical optimum remains fundamentally the same. Additionally, the OCL setting leads to some natural parallels with large scale federated learning, where there are a vast number of clients each with only a small number of datapoints.
>
> In this setting, it is typical that an individual client might only be accessed a few times in the overall training run, as FL systems tend to not get updates from every client per round in such large settings. This mirrors OCL both in data availability and also in privacy settings. The noise from differential privacy on the client side would completely drown out model updates, requiring more sophisticated privacy preserving strategies. OCL regimes where privacy preservation is required could benefit from reading research on privacy preservation in large scale FL systems.
>
> Regarding the 'Call to Action': This is an excellent suggestion. We are adding a dedicated “Call to Action” section. Our main call to action beyond “comparing notes” is understanding that if a researcher has published a method in a FL or CL setting, a next logical step is considering if that method functions in a more generalized partitioned learning setting. For example, existing work on (https://proceedings.mlr.press/v286/ceccherini25a.html)  uses SVD to guarantee stability in a CL settings with network data. Given the existence of Federated SVD (https://arxiv.org/abs/2105.08925) , a natural question could be, could one apply the same approach in a FL setting?
>
> Regarding Figure 5: Thank you for catching this. We have updated Figure 5 in the revision to explicitly label the Incompatibility Gap ($\Delta_z$) between the respective optimal parameter regions.

---

> > ### Author Rebuttal · Reviewer_Vg97 · 2026-04-03
> >
> > I thank the authors for the response. While the authors discuss on online continual learning, some of my main concerns regarding the fundamental differences between CL and FL and no concrete proposals still remain. I agree with reviewer Ki1r on this point. I maintain the same score.

---

### Decision · Program_Chairs · 2026-04-30

**Decision:**

Reject

**Comment:**

The observation is valid and interesting, but as the reviewers argued, is suffers from several limitations. First, while the problems have commonalities, they also have striking differences as the authors acknowledge, for example, in federated learning, privacy plays an important role while in continual learning it is not a central instrument. While these differences are acknowledged, the argument that despite these differences there is value in discussing the symmetry is valid but requires deeper discussion and argumentation. A second major limitation is the vagueness of the call for action. It is always beneficial to being familiar with broad literature, however, there is a lack of more specific path that will create value.